# Use of Potential Immobilized Enzymes for the Modification of Liquid Foods in the Food Industry

**Ernestina Garcia-Quinto [1], Raquel Aranda-Cañada [1], Paz García-García [2],* and Gloria Fernández-Lorente [1],***

[1] Laboratory of Microbiology and Food Biocatalysis, Institute of Food Science Research (CIAL, CSIC-UAM), Nicolás Cabrera, 9, UAM Campus, Cantoblanco, 28049 Madrid, Spain; ernestina.garcia@csic.es (E.G.-Q.); raquel.aranda@estudiante.uam.es (R.A.-C.)

[2] Healthy Lipids Group, Departmental Section of Food Sciences, Faculty of Sciences, Universidad Autónoma de Madrid, 28049 Madrid, Spain

* Correspondence: mariap.garcia@uam.es (P.G.-G.); g.f.lorente@csic.es (G.F.-L.)

**Abstract:** Enzymes are complex proteins that carry out biochemical reactions. Apart from being necessary for life, they are used in numerous industrial processes, especially in the textile, pharmaceutical, food and chemical sectors. One of the longest-lived industries regarding the use of enzymes is the food industry. Enzymes have always been used, mainly in their free form, to obtain new products and to improve the organoleptic qualities in different industries, such as in dairy, fruit and vegetables, and beverages. However, today, immobilized enzymes are the focus of attention in the liquid food industry, as they offer numerous advantages, such as stabilization and reuse, which enable cost reduction.

**Keywords:** enzymes; immobilized enzymes; immobilization; biocatalysts; food industry

## 1. Introduction

Enzymes are macromolecules with specific biochemical activities that have applications in numerous industries, such as the chemical, textile, food and pharmaceutical sectors. In the food industry, enzymes are used to recover by-products, facilitate manufacturing, improve aroma, and/or stabilize food quality. These macromolecules can catalyze the most complex chemical processes under the most favorable environmental and experimental conditions thanks to their excellent functional properties of activity, selectivity, and specificity [1]. Moreover, as they are generally considered as safe (GRAS) from a legal point of view, their use in food processing has been promoted [2].

The food industry can obtain enzymes from animal or plant tissues or via fermentation processes using selected microorganisms [3]. However, the use of enzymes is limited by availability and cost; therefore, microorganisms, as a source of enzymes, have become the most widely used option in recent years [4]. On the other hand, the immobilization of enzymes allows industries to simplify and reduce the cost of many industrial processes, as enzymes can be reused in industrial conditions [5]. Biocatalyst engineering, used for the preparation of highly efficient and very robust catalysts, is the key to process optimization and simplification. The combination of products of interest and optimized processes is crucial for novel foods to become an industrial and social reality. Thus, the development of simple protocols for the immobilization and stabilization of enzymes has become extremely important in the field of enzyme biotechnology applied to food technology [6].

This review summarizes the biotechnological state of the art regarding the enzymes applied in the food industry [7]. The choice between using enzymes either in their soluble or insoluble form will largely depend on the cost of the process. The use of immobilized enzymes for low-viscosity liquid-phase processes, such as for juices, brewing or dairy, seems to be the most interesting alternative with which to reduce costs, thanks to the advantages offered by the immobilization techniques. Liquid-phase processes are not limited by

mass transfer problems. Thus, some of the most commonly immobilized enzymes will be presented for application in different areas of the food industry.

## 2. Enzymes and Classification

Enzyme comes from a Latin word meaning "in yeast". Enzymes are biological protein macromolecules, produced by a living organism, that act as catalysts to trigger a specific biochemical reaction [4].

Regarding their properties, enzymes are selective catalysts that accelerate and increase the specificity of metabolic reactions [4,8]. Moreover, enzymes are selective for a particular substrate, and both the enzyme and the substrate have complementary geometric shapes; for this reason, they are nicknamed the "key and lock" model. Only one part of the enzyme, called the active center, is responsible for catalysis, and is usually around 2–4 amino acids. They also have areas in which cofactors bind, which are necessary for catalysis to take place [9,10].

As proteins, enzymes can be denatured under harsh conditions by heat or chemicals, leading to reversible or irreversible inactivation. Thus, each enzyme has its own optimal reaction condition [10].

Enzyme nomenclature, as well as classification, depends on the nature of the reaction, as shown in Table 1 [11,12]. The EC (Enzyme Commission) number is used to classify enzymes according to their chemical reaction in a numerical manner [11,13].

**Table 1.** Summary of enzyme classification according to the nature of the reaction, with the EC number and examples of enzymes in each group updated to the latest version of 2018 [11,12].

| Enzyme Type | EC | Reaction | Examples |
| --- | --- | --- | --- |
| Oxidoreductase | EC1 | Oxidation reactions involve the transfer of electrons from one molecule to another | Lipoxidases Dehydrogenases Glucose oxidase |
| Transferase | EC2 | They catalyze the transfer of groups of atoms from one molecule to another | Aminotransferase Transaminase |
| Hydrolase | EC3 | Hydrolysis reactions involve the cleavage of substrates by water | Lactase Proteases Trypsin |
| Lyase | EC4 | Catalyze the addition of groups to double bonds or the formation of double bonds via the removal of groups | Pectate lyases Decarboxylase Hydratases |
| Isomerase | EC5 | Catalyze the transfer of groups from one position to another on the same molecule | Topoisomerase Glucose isomerase |
| Ligase | EC6 | They catalyze the joining of two molecules to form a new bond | Glutathione synthase Aminoacyl tRNA synthetase |
| Translocases | EC 7 | Catalyze the movement of ions or molecules across membranes or their separation within membranes | Ubiquinone reductase ATP synthase Ascorbate ferrireductase |

## 3. Enzymes in Food Processing

Enzymes have always been of particular importance in the food industry. As early as 6000 BC, enzymes were used in the production of certain foods such as cheese, bread, beer, and wine. However, it was not until 1960 that they began to be used on a large scale [7,14,15]. Today, their use is prominent in the food industry, modifying and improving the nutritional, functional and sensory properties of both ingredients and products. Among the most

used enzymes are hydrolases, accounting for 75%, and the most important carbohydrases, lipases and proteases [7,15]. Below, some examples of enzymes in their soluble form that are commonly used in different processes in the food industry are detailed.

### 3.1. Enzymes in Fruit and Vegetable Industry

Fruits and vegetables are foods that can be eaten without processing, but different products can also be obtained from them after different treatments. In general, lignocellulolytic enzymes are used to hydrolyze lignocellulose. Lignocellulose refers to the plant biomass of woody and non-woody plants. It consists mostly of lignin, cellulose, hemicellulose and pectins. These enzymes degrade lignocellulosic substrates into glucose and other soluble sugars, which is why they are used to provide flavor and texture in the vegetable and fruit industry in order to produce foods such as juices, soups, and oils [16].

Vegetable oils such as coconut, olive and almond are extracted following a series of steps. The extraction is commonly performed via a chemical process using hexane, but it can also be developed biotechnologically using pectinolytic enzymes that degrade the cell wall and soften the structural components, allowing the quality of the oil to be improved by increasing the number of phenolic compounds [16,17]. Olive oil is considered the healthiest oil as it contains 71% monounsaturated fats. To avoid acidity, rancidity and the lack of aroma, an enzyme mixture consisting of cellulases and hemicellulases is also widely used to increase stability and improve its nutritional properties after extraction [18,19].

Inside the fruit and vegetable industry, juices are of high importance due to their nutritional characteristics. In this industry, the use of enzymes is superior to traditional mechanical and thermal processing as they are used to accelerate juice extraction, increase efficiency in pressing and sedimentation, and thus to produce clear and concentrated juices [15,16,20]. In this case, cellulases (endoglucanase and β-glucosidase), hemicellulases (endo- and exoxylanase and xyloglucanase) and pectinases (endo- and exo-arabinases) are used to aid technological processes, to improve organoleptic aspects and to reduce bitterness [21]. If fruits do not require peeling, extraction with pectinases is used directly, which facilitates the pressing and separation of the pulp either by filtration, sedimentation, or centrifugation. In the apple example, rhamnogalacturonase obtained from *Aspergillus aculeatus* is used to separate the pulp [21].

### 3.2. Enzymes in Brewing Industry

Although brewing beer dates back six thousand years, brewing is still booming today, as it is the most widely consumed alcoholic beverage in the world. Enzymes are key to brewing beer, both endogenous and exogenous [22,23].

The main raw material for beer is barley, which contains starch that is broken down into fermentable sugars that can then be fermented by yeast to produce ethyl alcohol [22]. The first step is malting, in which complex carbohydrates are converted into dextrins and maltose via the addition of α-amylases and proteases. The next step, which requires enzymes, is fermentation, leading to the production of ethanol and carbon dioxide [22,23].

Enzymes are used in the germination, mashing, fermentation, and clarification steps, so enzymes play a key role in brewing beer. In particular, the most important enzymes are β-glucanases, proteases, α-amylases, and beta-amylases. Other commercial enzymes are also often used to provide certain added values [22,23].

Even though beer has been produced for centuries, there are still innovations that can be made in this field, such as light beer, and the demand for this type of product has made it necessary to implement it in the brewing industry. One of the ways to obtain it is by adding glucosamylases to the wort before or during the fermentation process [24]. The first time a light beer was produced was in 1989 by Dr. Owades. Light beer contains 20% more fermentable carbohydrates, and this translates into one-third fewer calories compared to regular beer [24,25].

### 3.3. Enzymes in Dairy Industry

One of the first enzymes used in the dairy industry was used for cheese production using rennet. Rennet is a mixture of enzymes formed by chymosin and pepsin that act on the milk protein, producing its coagulation [15,26]. However, due to the need for calf rennet alternatives, many attempts have been made to find new proteases. In a recent study, a novel cysteine protease with milk-clotting activity was purified from *Ficus johannis* [27]. The new protease might be a promising candidate for the dairy industry, as well as other food and biotechnological industries.

Currently, one of the most widely used enzymes is β-galactosidase (also known as lactase), which can hydrolyze lactose into glucose and galactose, thus reducing the immunogenicity that can be produced in lactose intolerants [26,28,29]. Traditionally, neutral β-galactosidases have been obtained from lactic yeasts. Nowadays, genetically modified microorganisms (GMOs) are being used to produce β-galactosidases, thereby increasing the specific activity of the enzyme. However, the use of GMO enzymes has not been implemented yet [29]. The enzymes are administered in dairy products, which are kept under agitation and refrigerated temperatures; once lactose is degraded, the product is pasteurized or sterilized so that the enzyme is inactivated and no activity is present in the final product [26,29].

In a recent study, a GMO was used in the yogurt industry. Some researchers have developed a chemical gene synthesis strategy to obtain β-D-fructofuranosidase. This new gene was expressed in *Escherichia coli* to obtain the recombinant protein. The addition of this protein to yogurts enhanced the growth of the probiotics, thus increasing their nutritional value. In addition, this improved the formation of the yogurt gel, reducing the gel formation time and decreasing the elasticity while increasing its viscosity [30].

## 4. Immobilized Enzymes and Their Methods

Soluble enzymes, despite their excellent functional properties, have not been optimized to work in industrial reactors with high temperatures, an extreme pH, high substrate and product concentrations, mechanical agitation, or solvent use, as they may be compromised. The solution to these problems appears with enzyme engineering, which is considered to be one of the most exciting interdisciplinary and complex objectives of biotechnology [1], in which enzyme immobilization technologies are developed to improve enzyme properties in the food industry.

Immobilization techniques are defined as processes in which enzymes are localized into their insoluble forms, either using supports or not, retaining and in some cases improving their activity and stability. In this way, immobilized enzymes can be easily separated from the reaction media and their recyclability and reuse is promoted. Immobilized and stabilized enzymes could be reused for long periods of time under harsh industrial conditions, thus enabling an economically worthy process, which also offers many advantages, such as a simplified reactor design and reaction control [31].

Immobilization techniques could be classified into irreversible or reversible methods. Irreversible immobilization occurs when a biocatalyst binds covalently to a support and, unless the biological activity of the enzyme is destroyed, the binding cannot be separated. Moreover, irreversible immobilization could be developed via encapsulation or cross-linking without using supports. Support enzyme immobilization can be established via physical or chemical interactions. Therefore, in reversible immobilization, immobilized enzymes can be detached from the support under mild conditions [32]. The latter approach is very attractive, mainly for economic reasons, since when the enzyme activity decreases, the support can be reused and replenished with a new enzyme. As shown in Figure 1, adsorption and ionic interaction for binding to supports are among the reversible methods, while the rest are irreversible methods [33].

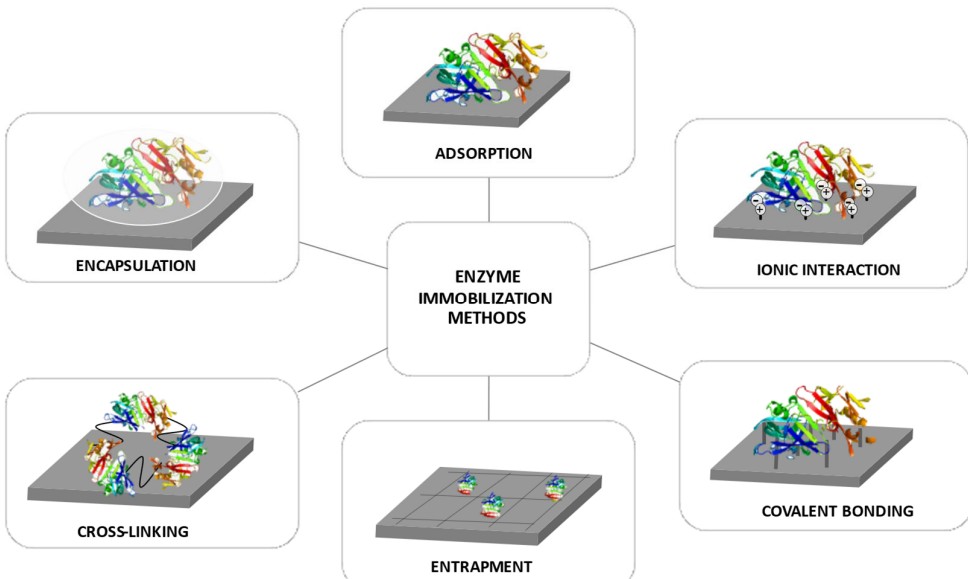

**Figure 1.** Major enzyme immobilization methods. Reversible methods: adsorption and ionic interaction. Irreversible methods: covalent bonding, entrapment, cross-linking and encapsulation.

When choosing the proper immobilization method for each intended application and enzyme, the following requirements must be considered [34]: (1) the preservation of enzyme activity by keeping it as close as to its native state as possible; (2) assurance of the stability of the biomolecule; (3) selectivity and sensitivity for the substrate of interest; (4) maximum enzyme activity per unit matrix; and (5) low cost.

Choosing the right type of material to use as a matrix, carrier, or support for enzyme immobilization is essential in order to achieve the maximum enzyme loading, which is the primary goal of any immobilization process in order to develop efficient and fast reactions.

The choice of support depends on certain factors that play an indispensable role in industrial applications (microbial resistance, mechanical resistance, chemical durability, thermal stability, etc.) [35]. Some of the most used organic supports are agarose, cellulose, starch, collodion, dextran, nylon, collagen, chitosan, and copolymers such as maleic anhydride and ethylene. In inorganic supports, we find materials such as colloidal silica, glass particles, alumina, nickel oxide, kaolinite, charcoal, and hydroxyapatite, among others [36,37]. Increasing numbers of new supports are being developed with better features.

Very often in industry, a continuous-flow operational configuration is preferred over batch processes, especially when large quantities of products are desired [5]. For this purpose, industry usually employs three types of reactors that use immobilized enzymes; small volume stirred-tank batch reactors, generally used in the pharmaceutical industry; packed-bed bioreactors (PBRs), used by the chemical and food industry where the volumes of products manufactured are very large; and finally, a modification of these, fluidized-bed reactors (FBRs), in which the immobilized enzyme particles are held in suspension by the upward flowing substrate stream [38].

As shown in Table 2, some examples of immobilized enzymes with the great potential to be used on a large scale in a continuous process for the production of liquid foods such as juices, beers, wines, milk, oils are highlighted and further discussed in the text.

**Table 2.** Use of immobilized enzymes for the industrial manufacture of liquid food ingredients. Main industrial applications in the fruit and vegetable, brewing, dairy and functional food industries.

| Application in the Food Industry | Enzyme | Support | Immobilization | Reference |
|---|---|---|---|---|
| Clarification of pomegranate juice | Protease + Pectinase (multienzymatic system) | Chitosan beads | Covalent | [39] |
| Juice processing | Pectinase | Calcium alginate beads | Ionic | [40] |
| Beer off-flavor prevention | α-acetolactate decarboxylase | Glutaraldehyde-activated chitosan beads | Covalent | [41] |
| Elimination of biogenic amines in wine | Amine oxidase + Catalase (multienzymatic system) | Glyoxyl–agarose 6BCL | Multipoint covalent | [42] |
| Lactose-free milk production | β-galactosidase | Magnetic chitosan microsphere | Cross-linking | [43] |
| Low-fat cheese production | Lipase | α-lac nanotubes | Cross-linking | [44] |
| Conjugated linoleic acid production | Isomerase + Lipase | D301R + IRA-93 | Covalent | [45] |
| Functional omega-3 ingredients | Phospholipase | Immobeads-C18 | Adsorption | [46] |

## 5. Immobilized Enzyme Technology for Food Application

Soluble enzymes usually have poor stability under process conditions and cannot be recovered [1,47]. In contrast, as described above, the use of immobilized enzymes solves these problems. In addition, immobilized biocatalysts work better in continuous processes at the industrial level [48]. Nowadays, for continuous processes, the fluidized-bed reactor is widely used when substrates contain suspended particles or are viscous [48,49]. Thus, food additive synthesis and food processing can be carried out in an easier and more efficient way.

However, most of the data related to the application of immobilized enzymes are limited to laboratory experiences due to their difficult implementation in the food industry. Mentioned below are some of the most interesting studies on new enzyme immobilization methods for liquid food modification in recent years.

### 5.1. Immobilized Enzymes in the Fruit and Vegetable Industry

Pectinolytic enzymes are used for juice clarification with the aim of eliminating the turbidity caused by the presence of pectic substances and starch. Despite the superior catalytic activity of pectinase, the use of this enzyme in its free form has drawbacks, such as a lower stability under operating conditions, the inability to isolate the product and the inability to be continuously repeated in an industrial process [50].

As previously explained, these problems can be overcome by immobilizing enzymes, which increases catalyst stability and aids in continuous catalyst reuse [51]. In this case, the application of immobilized pectinase in juice clarification as an alternative to conventional processes is discussed.

A study conducted by Diano et al. [52] analyzed a series of pectic enzymes for apple juice clarification, which were covalently immobilized on three different types of supports: glass beads, PAN-beads (polyacrylonitrile based beads) and nylon 6/6 granules. They observed that when using a fluidized bioreactor with immobilized enzymes, complete pectin hydrolysis was achieved in 41 min, while 131 min was necessary with packed reactors [52]. As it can be seen in this article, FBRs for continuous juice clarification are a technology with great potential for industrial application, since they contribute to reducing production costs [53].

In a similar study, Benucci et al. [39] performed the covalent immobilization of pineapple stem bromelain (protease) and Pectinex® BE XXL (pectinase) on chitosan beads for the clarification of pomegranate juice. They applied a new multienzymatic system using a FBR, with which they obtained promising results [39]. Recently, in another study, an enzyme cocktail composed of pectinolytic and cellulolytic enzymes was immobilized on glutaraldehyde particles for orange juice clarification using a FBR, as seen in Figure 2. In this work, 0.22 g of biocatalyst was used to clarify more than 2 L of juice and, even after 3 days of operation, these biocatalysts were still pre-sensing 60% of their clarification capacity. Again, the use of FBR with small amounts of immobilized enzymes enabled greater operational stability compared to PBRs [54].

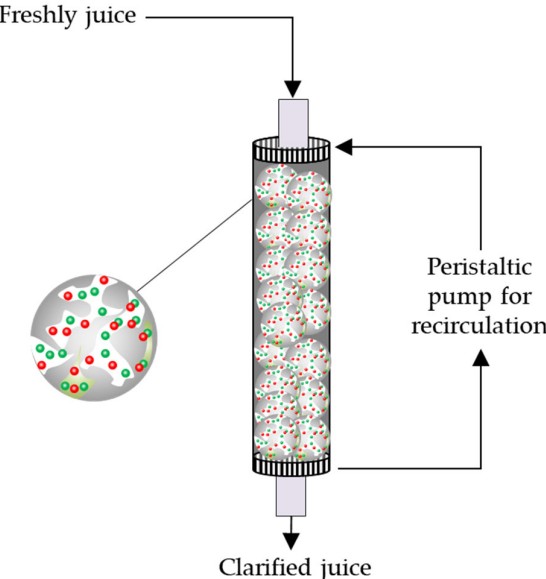

**Figure 2.** Flow diagram of the experimental equipment of the fluidized bed reactor (FBR) with immobilized enzyme cocktail for the clarification of juices in the food industry.

On the other hand, an active pectinase (PNs31) was purified from a *Bacillus* strain, and its biochemical characteristics were investigated. In addition, the immobilization of PNs31 with calcium alginate beads was carried out for the design of economic processes in industrial applications [40]. To our knowledge, PNs31 of *Bacillus subtilis* CBS31, with a unique amino acid sequence and bio characteristics, as well as approaches to a cost-effective method for enzyme immobilization, have never been reported.

In all the works mentioned, immobilized pectinase makes the enzyme less vulnerable to adverse conditions, prevents product contamination, improves process control, and enables the use of FBRs for high volumes of product [39].

### 5.2. Immobilized Enzymes in the Brewing Industry

The demand for gluten-free beer has strongly increased in recent years and its production has become an important socioeconomic issue. Therefore, AN-PEP (*Aspergillus niger* derived endopeptidase) immobilization has become one of the main targets of the brewing industry for the continuous production of gluten-reduced beer.

Benucci et al. [49] have demonstrated that AN-PEP immobilized on chitosan beads could be applied for the industrial production of gluten-reduced beer. Using this biocatalyst in an FBR, they reduced the gluten content in commercial beer to the requested values below 20 mg/kg [49]. Similarly, the covalent immobilization of α-acetolactate decarboxylase (ALDC) on glutaraldehyde-activated chitosan beads has recently been achieved for application in beers to prevent off-flavor [41]. In this study, the immobilized enzyme showed a stabilization factor almost seven times higher than that of a free-form enzyme, at 60 °C.

In relation to wine, hen egg white lysozyme (HEWL) has also been immobilized on microbial chitosan beads for the control of malolactic fermentation [55]. As it can be assumed from these examples, chitosan, used as a support, has attracted great attention due to its biodegradability, biocompatibility, non-toxicity, renewability, flexibility, and availability at many reactive sites, which provide important functions for multifunctional applications [56]. On the other hand, the co-immobilization and co-localization of two enzymes (amine oxidase and catalase) on the same porous support were achieved in order to instantly remove the hydrogen peroxide produced in the process of biogenic amine oxidation in wine. The co-localized optimal derivative promoted the instantaneous removal of 91% of the $H_2O_2$ released inside the porous support during putrescine oxidation. This optimal derivative retained 92% of the activity after three reaction cycles, whereas the immobilized amine oxidase without catalase retained only 41% of the activity [42]. From the above results, they concluded that the co-immobilization and co-localization of both enzymes within a porous structure were strictly necessary, processes that had never been studied and optimized for wine. In the near future, these active and stable catalytic preparations could be used for the design of new biosensors for the detection and elimination of biogenic amines, which can cause important intoxications in society [42,57,58].

### 5.3. Immobilized Enzymes in Dairy Industry

Several enzymes, including β-galactosidase, catalase, pepsin, and peroxidases, have been successfully studied regarding the ability of immobilization to increase the efficiency of their application in the processing of different dairy products. The enzyme β-galactosidase is mainly used to process dairy products and treat lactose intolerance, a condition in which the small intestine cannot digest or break down all the lactose ingested [59].

Ke et al. [43] synthesized a novel magnetic chitosan microsphere (MCM) as a carrier for lactase immobilization. Compared to free enzymes, the immobilized enzyme showed higher thermal stability, activity, and a better pH range [43]. Therefore, the immobilized enzyme would be used as a cheaper alternative in order to produce lactose-free milk. In another recent study, a magnetic graphene oxide nanocomposite (mGPP) was prepared as a vehicle for lactase immobilization. This mGPP–Lactase could be rapidly recycled via magnetic separation from the aqueous solution, maintaining 83.1% residual activity even after 20 consecutive cycles [60]. Both MCM and mGPP would be great candidates for future biomedical applications for cyclic lactose hydrolysis.

Referring to the cheese-making process, rennet and lipases are essential for the development of a characteristic flavor [61]. Narwal et al. [62] immobilized a milk coagulation enzyme (MCE) on an alginate–pectate interwoven gel with a yield of 73%. They obtained very good results with such a derivative, which showed high operational stability and retained 40% activity even after 10 uses [62]. Moreover, Memarpoor-Yazdi et al. [63] demonstrated that GDSL lipase from *Rhodothermus marinus* covalently immobilized on chitosan-coated $Fe_3O_4$ nanoparticles exhibits the high hydrolytic capacity of short-chain esters, suggesting its potential to improve cheese flavors. In another interesting study, a new lipase immobilization vehicle using self-assembled α-lac nanotubes enhanced their catalytic activity to create low-fat cheeses with improved flavor [44]. Finally, in a recent study, the immobilization of the probiotic *Lacticaseibacillus paracasei* KC39 on wheat bran was investigated as a vehicle to produce soft cheese and other functional products, which are on the rise [64].

### 5.4. Immobilized Enzymes in the Functional Food Industry

In recent years, the use of immobilized enzymes in the manufacture of nutraceuticals, food ingredients with beneficial health effects, has increased due to the great interest in these new products [65]. Immobilization technology is used to isolate nutraceuticals and to incorporate them into common foods to enhance their added value [3].

On one hand, immobilized lipase preparations from *Candida antarctica* and *Rhizomucor miehei* have demonstrated the ability to increase the conjugated linoleic acid (CLA) con-

tent, a very healthy polyunsaturated fatty acid from the Omega-6 group, of the milk fat acylglycerols. Comparable increases have also been obtained using a free enzyme from *Candida rugosa* [66]. On the other hand, a novel and efficient method has been developed for converting plant oil into a specific CLA (conversion ratio was 90.5%) using a synergistic biocatalytic system based on immobilized *Propionibacterium acnes* isomerase (PAI) and *Rhizopus oryzae* lipase (ROL) [45].

Omega-3 is another potential functional ingredient. Omega-3 fatty acid consumption, particularly that of eicosapentaenoic acid (EPA) and docosahexaenoic acid (DHA), has been associated with a decreased risk of cardiovascular, neurological and inflammatory-mediated diseases, as well as with the regulation of cellular activity [67]. Akanbi and Barrow [68] demonstrated that the fatty acid selectivity of *Candida antarctica* lipase A (CAL-A) immobilized on octadecyl-activated resin was applied to produce DHA concentrate by controlling the rate and extent of hydrolysis. In another study, the selectivity of different immobilized lipases, including CAL-B, *Thermomyces lanuginosus* (TLL) and *Rhizomucor miehei* (RML), for the ethanolysis of sardine oil to produce ethyl esters highly enriched in omega-3 fatty acids was studied [69,70]. All these processes had the great advantage of being able to be repeated by using immobilized enzymes.

Normally, to enrich the omega-3 polyunsaturated fatty acid level in foods, EPA and DHA from fish oils are purified and stabilized as ethyl esters. However, triacylglycerols are the most desirable molecular form as they are most rapidly metabolized. The synthesis of structured triglycerides with DHA esterified in all sn-positions has been described in the literature, reaching a product yield higher than 80% using commercial lipases adsorbed on hydrophobic supports [71]. Other than that, lysophospholipids (LPL) are considered as important bioactive lipids because many of them are involved in a wide variety of pathological processes [72]. Garcia-Quinto et al. [46] produced two phospholipids in mono- and di-substituted forms using different lipases and phospholipases immobilized via interfacial adsorption on hydrophobic supports. In the work, they studied different reaction parameters, such as the increase in the solubility of the limiting substrate and the selectivity of the immobilized enzymes, to produce interesting functional ingredients [46]. In the future, the costs generated from this production will be economically viable thanks to the immobilization and stabilization of enzymes.

## 6. Conclusions

Overall, this review covers recent studies on both soluble and immobilized enzymes in the food industry in recent years. First, the various classifications of the types of enzymes and their most used functions in different areas of the food industry are explained. Then, the enzyme immobilization technique is introduced as a solution to the problems of cost and the unstable nature of enzymes that prevent their industrial application. Within this, enzyme–support interactions are explained and the supports that exist nowadays are detailed, as well as the most used reactors in the food industry. Finally, new enzyme immobilization methods that are carried out for different types of food industries are discussed.

The choice of enzyme type, support material, immobilization method and application in the food industry influences the overall economics of the process, which is a key factor in successfully commercializing an immobilized enzyme system. Soon, immobilized enzymes will play a very important role in the food industry, as their application will reduce overall process costs due to their high reusability. Furthermore, immobilized food enzymes improve catalytic activity, pH and thermal stability, and the operational capacity in a way that free enzymes do not. Therefore, the development and optimization of new immobilization strategies in order to produce highly stable biocatalysts will overcome the potential drawbacks associated with industrial scale-up.

**Author Contributions:** Conceptualization: P.G.-G., E.G.-Q. and R.A.-C.; Writing—original draft preparation: E.G.-Q. and R.A.-C.; Writing—review and editing: P.G.-G. and G.F.-L.; Supervision: P.G.-G.; Project administration: G.F.-L.; Funding acquisition: G.F.-L. All authors have read and agreed to the published version of the manuscript.

**Funding:** The authors gratefully acknowledge the financial support from the Ministry of Science and Innovation, Spain (Number project No. RTI2018-093583-B-I00) and project ALGATEC-CM (P2018/BAA-4532), supported by the Comunidad de Madrid (Spain) and co-financed by the European Social Fund.

**Acknowledgments:** The authors would like to thank the Industrial Doctorate Programme (IND2022/BIO-23509) granted to Raquel Aranda-Cañada and funded by Comunidad de Madrid (Spain).

**Conflicts of Interest:** The authors declare no conflict of interest.

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
