# Peer review of "Use of Potential Immobilized Enzymes for the Modification of Liquid Foods in the Food Industry"

_processes, doi:10.3390/pr11061840_

Round 1

Reviewer 1 Report

The presented manuscript is an attempt to review enzymes used in the food industry in both soluble and immobilized forms, with an emphasis on the advantages of the immobilized form. The scope of the review covered several food production industries, such as the dairy, baking, brewing, meat, and fruit and vegetable industries.

The idea of the article is interesting, but the implemented scope, content and form are not fully satisfactory. In reviewer’s opinion, the manuscript needs substantial improvement as noted below. In conclusion, I recommend this manuscript for publication after major revision.

Comments:

1.     In recent years, there have been quite a few review articles on enzymes as well as the immobilization of enzymes used in the food industry. A few of them were used by the authors and cited, but there are still a lot of available review items that you can use, such as: doi:10.1021/acsomega.2c07560, doi:10.1080/10408398.2022.2092719, doi.org/10.1016/j.cofs.2021.09.004, doi.org/10.1590/fst.106222, doi.org/10.1016/B978-0-12-813280-7.00041-4, doi.org/10.1016/B978-0-12-813280-7.00037-2, doi.org/10.20431/2349-0403.0702002.

2.     Other review articles on enzymes in the food industry are conducted in different perspectives, areas and forms. Many of them are of a very wide scope, in a specific approach or a unique form of presentation and ordering criterion. Authors need to think carefully about how they want to achieve the uniqueness of their review. In my opinion, its current form and content are too concise, superficial and unremarkable.

3.     In my opinion, the title is ambiguous. Please consider clarifying it.

4.     It would be good to add some lines in the tables 1 and 2 to improve their readability. In addition, an increase in the number of tables in the manuscript could be attractive to the reader, in which the previous knowledge of various researchers could be clearly synthesized and systematized with references.

5.     In sections 3 and 5, the subsections should be in the same order (according to the industries discussed). In addition, these sections should be expanded to include more examples from a given industry. In the recent literature, you can find more articles dealing with enzymes in the specific food industry (for example, in the meat industry: doi.org/10.3390/foods12061336).

You can also extend the review to other food sectors (e.g. wine, sugar, cocoa, etc.).

6.     Citations: items 9 and 10 - renumbering (reverse order of appearance in the text), item 48 - no citations in the text. Also, check citations in the text with the authors' names: Ke et al., 2020 (line 270), Rajesh et al., 2016 (line 279).

In addition, please check all citations to comply with the ‘Reference List and Citations Style Guide for MDPI Journals’. For example, instead of '[6], [7]' it should be '[6-7]' or instead ‘Benucci et al., 2019 showed … [49]’ it should be ‘Benucci et al. [49] showed …’.

7.     References - please carefully check the journal's requirements for the description of references and correct the entire reference list.

Items 1, 10, 15, 29, 41, 49 do not specify the year of publication.

8.     There are editing errors in several places (no spaces, unnecessary spaces, no periods - lines: 11, 12, 45, 68, 68, 77, 87, 114, 258, ) and in two places an unfinished sentence (lines 51, 79).

Author Response

In recent years, there have been quite a few review articles on enzymes as well as the immobilization of enzymes used in the food industry. A few of them were used by the authors and cited, but there are still a lot of available review items that you can use, such as: doi:10.1021/acsomega.2c07560, doi:10.1080/10408398.2022.2092719, doi.org/10.1016/j.cofs.2021.09.004, doi.org/10.1590/fst.106222, doi.org/10.1016/B978-0-12-813280-7.00041-4, doi.org/10.1016/B978-0-12-813280-7.00037-2, doi.org/10.20431/2349-0403.0702002. 1.

More references are included with recent dates of publication.

  1. Other review articles on enzymes in the food industry are conducted in different perspectives, areas and forms. Many of them are of a very wide scope, in a specific approach or a unique form of presentation and ordering criterion. Authors need to think carefully about how they want to achieve the uniqueness of their review. In my opinion, its current form and content are too concise, superficial and unremarkable.

This review is focused on the advantages that immobilized enzymes have in food industry compared to soluble enzymes, and specifically in liquid food industry such as fruits and vegetables, dairy, brewing and functional ingredients.

  1. In my opinion, the title is ambiguous. Please consider clarifying it.

The title has been modified: “Use of potential immobilized enzymes for the modification of liquid foods in the food industry”.

  1. It would be good to add some lines in the tables 1 and 2 to improve their readability. In addition, an increase in the number of tables in the manuscript could be attractive to the reader, in which the previous knowledge of various researchers could be clearly synthesized and systematized with references. 4.

Thank you for your comments. Improvement in tables and the addition of figures were included.

  1. In sections 3 and 5, the subsections should be in the same order (according to the industries discussed). In addition, these sections should be expanded to include more examples from a given industry. In the recent literature, you can find more articles dealing with enzymes in the specific food industry (for example, in the meat industry: doi.org/10.3390/foods12061336).

You can also extend the review to other food sectors (e.g. wine, sugar, cocoa, etc.).

Meat industry has been removed in order to focus the review on liquid food industry. Wine examples are also added to the review. Functional ingredients in which modification of oils is also added.

  1. Citations: items 9 and 10 - renumbering (reverse order of appearance in the text), item 48 - no citations in the text. Also, check citations in the text with the authors' names: Ke et al., 2020 (line 270), Rajesh et al., 2016 (line 279). In addition, please check all citations to comply with the ‘Reference List and Citations Style Guide for MDPI Journals’. For example, instead of '[6], [7]' it should be '[6-7]' or instead ‘Benucci et al., 2019 showed … [49]’ it should be ‘Benucci et al. [49] showed …

Citations errors have been solved.

  1. References - please carefully check the journal's requirements for the description of references and correct the entire reference list. Items 1, 10, 15, 29, 41, 49 do not specify the year of publication.

References have been modified.

  1. There are editing errors in several places (no spaces, unnecessary spaces, no periods - lines: 11, 12, 45, 68, 68, 77, 87, 114, 258, ) and in two places an unfinished sentence (lines 51, 79).

Thank you for your comments, errors solved.

Reviewer 2 Report

In this review, the authors focus on the enzymes which have been used in the food industry. The enzyme classification and the enzymes included in dairy, baking, fruit and vegetable, brewing, and meat industry were introduced. And the immobilization of the enzymes was also presented. However, there are some issues that should be solved before publication, and the detailed comments were listed below:

1) There are 7 enzyme types. Translocases were added as EC 7 by IUBMB in 2018.

2) There is no figure in the whole manuscript. It would be interesting for the readers if the authors could add the catalytic process or mechanism in the manuscript.

3) The organization of the manuscript is poor. For example, there are only two enzymes mentioned in the dairy industry and no detailed discussion.

4) The immobilization of the enzymes in the manuscript could be integrated into the applications in the food industry.

5) There are too many errors. For example, some spaces are missing: line 11, ......line 79, remove the "the".

The spelling and the presentation should be improved.

Author Response

  • There are 7 enzyme types. Translocases were added as EC 7 by IUBMB in 2018.

Added in Table 1.

  • There is no figure in the whole manuscript. It would be interesting for the readers if the authors could add the catalytic process or mechanism in the manuscript.

Types of immobilizations are included in form of figure instead of the table. Moreover, a figure regarding the schematic composition of fluidized bed bioreactors (FBR) are included as it is one of the most used reactors in liquid food industry such as modification of juices.

  • The organization of the manuscript is poor. For example, there are only two enzymes mentioned in the dairy industry and no detailed discussion.

More examples are added and discussed.

  • The immobilization of the enzymes in the manuscript could be integrated into the applications in the food industry.

Importance of immobilized enzymes is highlighted in the review. Thus, different applications in which immobilized enzymes are better than soluble enzymes in liquid food industry are included.

  • There are too many errors. For example, some spaces are missing: line 11, ......line 79, remove the "the".

Done. 

Round 2

Reviewer 1 Report

The authors significantly improved the manuscript, revised the scope of the review, and incorporated most of my comments. I recommend the manuscript for publication in its present form.

I raise only one point for the authors' and editors' consideration: a slight change of title. In my opinion, the fragment '... liquid foods in food industry' is a bit clumsy. I would suggest changing it to, for example, '...liquid foods' or '...liquid foods in industrial processes' or otherwise.

Reviewer 2 Report

After the modification, it could be accepted for publication in the present form.

The English has been improved in the revised version.